# Dynamic influence of pinning points on marine ice-sheet stability:
## *a numerical study in Dronning Maud Land, East Antarctica*

Lionel Favier[1], Frank Pattyn[1], Sophie Berger[1], and Reinhard Drews[1]

[1]Laboratoire de Glaciologie, DGES, Université libre de Bruxelles, Brussels, Belgium

*Correspondence to:* Lionel Favier (lionel.favier@ulb.ac.be)

**Abstract.**

The East Antarctic ice sheet is likely more stable than its West Antarctic counterpart, because its bed is largely lying above sea level. However, the ice sheet in Dronning Maud Land, East Antarctica, contains marine sectors that are in contact with the ocean through overdeepened marine basins interspersed by grounded ice promontories and ice rises, pinning and stabilising the ice shelves. In this paper, we use the ice-sheet model BISICLES to investigate the effect of sub-ice shelf melting, using a series of scenarios compliant with current values, on the ice-dynamic stability of the outlet glaciers between the Lazarev and Roi Baudouin ice shelves over the next millennium. Overall, the sub-ice shelf melting substantially impacts the sea-level contribution. Locally, we predict a short-term rapid grounding-line retreat of the overdeepened outlet glacier Hansenbreen, which further induces the transition of the bordering ice promontories into ice rises. Furthermore, our analysis demonstrates that the onset of the marine ice-sheet retreat and subsequent promontory transition into ice rise is controlled by small pinning points, mostly uncharted in pan-Antarctic datasets. Pinning points have a twofold impact on marine ice sheets. They decrease the ice discharge by buttressing effect, and play a crucial role in initialising marine ice sheets through data assimilation, leading to errors in ice-shelf rheology when omitted. Our results show that unpinning increases the sea level rise by 10% while omitting the same pinning point in data assimilation decreases it by 10%, but the more striking effect is in the promontory transition time, advanced by two centuries for unpinning and delayed by almost half a millennium when the pinning point is missing in data assimilation. Pinning points exert a subtle influence on ice dynamics at the kilometre scale, which calls for a better knowledge of the Antarctic margins.

## 1  Introduction

The marine ice-sheet instability (MISI) (Weertman, 1974; Thomas and Bentley, 1978) hypothesis states that a marine ice sheet having its grounding line - the boundary between grounded and floating ice - resting on an upsloping bed towards the sea is potentially unstable. A prior retreat of the grounding line (e.g. ocean driven) resting on such an upsloping bed thickens the ice at the grounding line, which increases the ice flux and induces further retreat, etc., until a downsloping bed is reached, provided that upstream snowfall balances local flux at the grounding line. The MISI hypothesis has been verified using the boundary layer theory developed by Schoof (2007) and simulated with numerical studies (e.g. Durand et al., 2009). Because most of the West Antarctic ice sheet (WAIS) rests on an upsloping bed, the potential retreat of its grounding line is widespread. Therefore,

the vulnerability of the WAIS to current climate change has been extensively studied (e.g. Cornford et al., 2015; Deconto and Pollard, 2016). On the other hand, the East Antarctic ice sheet (EAIS) is less vulnerable to retreat on short time scales and its stability has therefore been less debated. However, a recent numerical study investigating ice-sheet instability in Antarctica through a statistical approach (Ritz et al., 2015) pointed out the likeliness for unstable retreat of grounding lines in Dronning Maud Land (DML), East Antarctica, over the next two centuries. Moreover, the EAIS hosts ten times more ice than the WAIS and, therefore its future stability needs to be more investigated.

In DML, the floating margins of outlet glaciers are buttressed by numerous topographic highs, which attach to the otherwise floating ice shelves from beneath and form icy pinning points protruding through ice. Pinning points are either called ice rises or ice rumples. The former exhibit a local-flow regime while the latter are still overridden by the main ice flow. Ice promontories are ice rises that are connected to the mainland through a grounded saddle. Most ice rumples and a significant number of ice rises are smaller than 10 km$^2$ (Matsuoka et al., 2015). Even though they are common features in DML, they are often missing in the bathymetry because airborne radar data are not closely enough spaced. This issue was recently pointed out in two studies revealing a series of uncharted pinning points from ice-sheet modelling (Fürst et al., 2015) and observations (Berger et al., 2016).

The back stress induced by pinning points - even small ones, i.e., few km$^2$ in area - buttresses ice shelves, hampering ice discharge towards the ocean. Because simulating pinning points requires accurate treatment of grounding-line dynamics, they have only recently been considered in ice-sheet models: Goldberg et al. (2009) and Favier et al. (2012) investigated the transient effect of pinning points for idealised geometry, using ice-sheet models of varying complexity. In both studies, including a pinning point beneath an ice shelf in steady state significantly slows down the ice flow, inducing a grounding-line advance until the grounded ice sheet fully covers the pinning point. The development of an ice rise over a deglaciation and its further stability among an ice sheet/shelf system in steady state was lately simulated by Favier and Pattyn (2015), even though the stability of ice rises has been known for decades (Raymond, 1983). Favier and Pattyn (2015) also demonstrated that ice promontories are transient features transitioning into ice rises during ice-sheet deglaciation.

Both studies of Favier et al. (2012) and Favier and Pattyn (2015) used ice-sheet models of sufficient complexity to accurately quantify the stress pattern in the pinning-point's vicinity: ice is compressed along flow upstream of the pinning point, sheared when flowing around it, and stretched along flow farther downstream. The levels of extensive stress computed were higher than what can be accommodated by ice creep, which in reality leads to brittle fracturing and rifting (Humbert and Steinhage, 2011). Pinning points thus affect ice rheology by increasing local-scale deformability, which further impacts surface velocities.

Initialisation of transient simulations relies on data assimilation methods (e.g. MacAyeal, 1993). These are applied to observed ice geometry and surface velocity to infer poorly known parameters such as basal friction and ice stiffening/softening, the latter mostly accounting for crevasse-weakening and ice anisotropy. These parameters are inferred by minimising a cost function, which sums the mismatch between observed and modelled surface velocities and Tikhonov regularisation terms for each inferred parameter, the latter terms being tuned to provide continuous fields and avoid over fitting. Even though regularisation remains subjective, a sound trade-off between reducing velocity mismatch and over fitting can be achieved using the L-curve method (e.g. Morlighem et al., 2013; Jay-Allemand et al., 2011).

In areas where ice/bed geometry and surface velocity are not correctly resolved, the inferred parameters are likely flawed. Recently, Fürst et al. (2016) investigated the band of floating ice that can safely calve off without increasing ice discharge to the ocean. This result stems from a static analysis of the force balance between ocean pressure and ice internal stress state, which can flaw further transient simulations if pinning points are not accounted for (Fürst et al., 2015). Berger et al. (2016)

demonstrated through a diagnostic study that omitting the contact between a topographic high and the ice-shelf base during data assimilation yields excessive ice-shelf stiffening, which compensates for the lack of basal friction in order to match observed surface velocities. However, it remains unclear how such erroneous initialisation impacts the transient behaviour of the ice-sheet/shelf system, which is a question we address here.

Unpinning may occur over various time scales due to progressive ice-shelf thinning (Paolo et al., 2015; Pritchard et al., 2012),

erosion, rising sea level, tidal uplift (Schmeltz et al., 2001), or through the developments of rifts (Humbert and Steinhage, 2011). However, unpinning of Antarctic ice shelves has been poorly documented so far. According to Mouginot et al. (2014), the best explanation for the acceleration of the eastern ice shelf of Thwaites Glacier in the Amundsen sea sector since 2008 might be reduced buttressing from the pinning point at its terminus (also hypothesised in Tinto and Bell, 2011), even though other mechanisms such as sub-ice shelf melting may also be at play. In Larsen C ice shelf, the unpinning of the Bawden and Gipps ice

rises was simulated diagnostically (i.e., without ice geometry changes) by manually decreasing the basal drag (Borstad et al., 2013), which substantially accelerated the ice flow by up to 200 m a$^{-1}$ over an extent of about 100 km upstream. However, the transient evolution of ice geometry and velocity after unpinning has not been investigated so far. We also address this question in this paper.

The studied area is situated between the Lazarev and Roi Baudouin ice shelves in DML and contains a number of ice

streams flowing around the Sør Rondane mountain range to the west and the Yamato mountain range to the east. The coastal belt comprises a series of ice rumples, ice rises and promontories buttressing the ice shelves. From west to east, the three outlet glaciers of Tussebreen (TB), Hansenbreen (HB) and West Ragnhild (WRG) are potentially unstable because their beds lie below sea level and dip towards the interior of the ice sheet. The grounded area is well constrained in the Antarctic-wide bed elevation datasets (Fretwell et al., 2013) as the latter incorporate airborne radio-echo sounding data collected during the

Austral summer of 2010/2011 (Callens et al., 2014, 2015). TB and HB are separated by the TB/HB promontory, HB and WRG by the HB/WRG promontory. The calving front of HB is in contact with two pinning points, hereafter called PPhs, and the calving front of WRG with another pinning point, hereafter called PPw (Figure 1).

The pinning point PPw strongly buttresses the ice shelf of WRG (Berger et al., 2016). However, Antarctic-wide datasets do not correctly resolve surface velocities (Rignot et al., 2011) and ice/bed geometry (Fretwell et al., 2013) in the vicinity of PPw.

This has been improved by Berger et al. (2016) who modified the corresponding datasets in the surroundings of PPw with field-based data of ice thickness and velocity (Drews, 2015). The modified datasets are used here for model initialisation.

In this study, we use the adaptive-mesh ice-sheet model BISICLES to investigate: (i) the future behaviour of these outlet glaciers (see previous paragraph) with respect to potential unstabilities, (ii) their dynamic response to PPw unpinning, (iii) the dependency of the transient results on the model initialisation, using datasets either resolving PPw (Berger et al. (2016)'s

high-resolution dataset), or not correctly resolving PPw (Rignot et al. (2011)'s velocities in combination with ice/bed geometry

from Fretwell et al. (2013)), and (iv) the effect of two sliding laws and six sub-ice shelf melting parametrisations comparable to observed values. The three distinct initial conditions stemming from (ii) and (iii) are used to run transient simulations forced by the different melting parametrisations, over the next millennium. The 36 resulting simulations give a comprehensive overview of future ice dynamics in DML and testify to the importance of including even small pinning points in the observational
datasets.

## 2 Datasets and Methods

### 2.1 Input data

Each experiment consists of an initialisation by data assimilation and a subsequent set of transient simulations. The initialisation requires surface velocity, ice thickness, bed elevation, englacial temperatures, and two initial fields for ice stiffening factor and
basal friction coefficient. The Transient simulations require ice thickness, bed elevation, initial englacial temperatures, two fields for ice stiffening factor and basal friction coefficient (the latter two computed by the data assimilation), surface mass balance, and basal mass balance of the ice shelves.

The computational domain covers an area of about 40,000 km$^2$ and is illustrated in Figure 1. Two distinct datasets for flow-field and ice/bed geometry were employed. The *standard* dataset comprises surface velocities from Rignot et al. (2011)
and ice/bed geometry from Fretwell et al. (2013) (the Bedmap2 dataset). The *high-resolution* dataset uses the observations of Berger et al. (2016) on the WRG ice shelf, which account for PPw in both surface velocities and ice/bed geometry (the latter called mBedmap2). These two datasets only differ for the WRG ice shelf and are otherwise identical.

Modelling grounding-line advance as a response to ocean-induced perturbation is very sensitive to sub-ice shelf bathymetry, which is roughly interpolated in our studied domain (Le Brocq et al., 2010) and thus largely uncertain. As a consequence, the
water column beneath ice shelves is in places very shallow, which can cause spurious ice-shelf re-grounding. In order to make the bathymetry more coherent with both bed elevation at the grounding line and (unpublished) measurements near the ice-shelf front, we lowered the bed elevation beneath the ice shelves in a two-step procedure. First, we excavated a 250 m thick uniform layer 30 km away from the grounding line, ensuring a smooth connection with the grounded area with a 1-D Gaussian function. The second part of the excavation is based on unpublished bathymetric data collected during a 2011 oceanographic survey (K.
Leonard, personal communication, 2012), which shows a more than 850 m deep trough cutting through the continental shelf between PPw and Derwael Ice Rise (DIR) (Figure 1). This feature may be the relict of past ice sheet erosion from the WRG ice stream when the grounding line was closer to the continental shelf break (Livingstone et al., 2012). We therefore assume the presence of a narrow trough cutting through the bathymetry beneath the ice shelf linked to the deepest section at the grounding line (yellow line in Figure 1). The second excavation was done across-flow using a 1-D Gaussian-shaped function (its half-
width is 15 km based on the ice-stream cross-section extent). Both excavations are included in the *standard* as well as the *high-resolution* datasets.

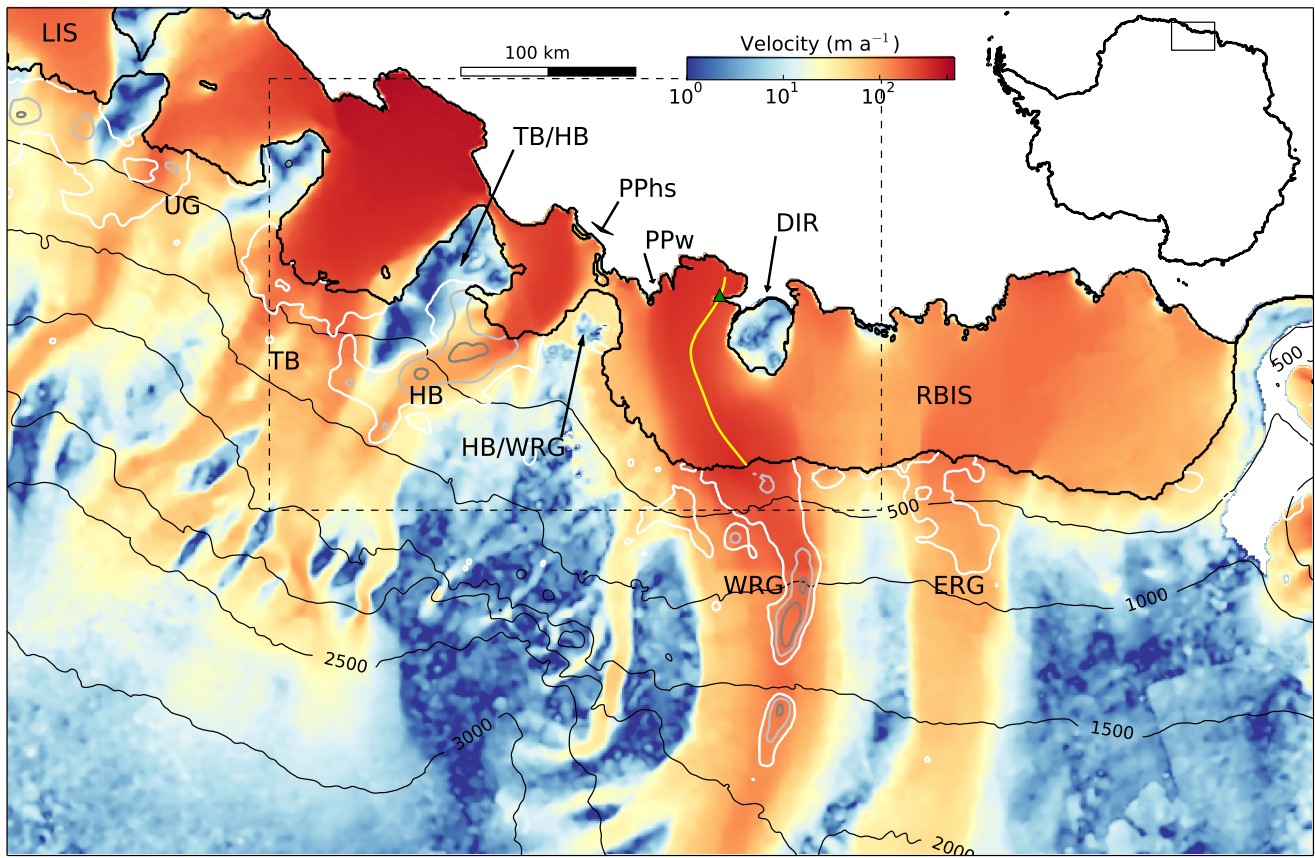

**Figure 1.** Computational domain with the extended flow-field from Berger et al. (2016) in the background. The thick black lines show the grounding line and the calving front. The thin black lines show ice surface elevation contours every 500 m. The white, light grey and dark grey lines are bed elevation contours of -500 m, -750 m and -1000 m, respectively. The yellow line shows the central trench of the bathymetry excavation (Section 2.1), and the green triangle the supporting bathymetric data (K. Leonard, personal communication, 2012). The dashed box shows the domain of interest shown from Figure 2 to Figure 7, and in Supplementary Figures. LIS: Lazarev Ice Shelf; UG: Unnamed Glacier; TB: Tussebreen; HB: Hansenbreen; HB/WRG: promontory in between HB and WRG; TB/HB: promontory in between TB and HB; DIR: Derwael Ice Rise; RBIS: Roi Baudouin Ice Shelf; WRG: West Ragnhild Glacier; ERG: East Ragnhild Glacier. We also name a group of two pinning points PPhs located at the front of HB, and the pinning point PPw at the front of WRG.

The surface mass balance was derived by Arthern et al. (2006), who combined in-situ measurements (most of them between 1950 and 1990) and satellite observations of passive microwave (from 1982 to 1997) using a geostatistical approach, and is constant in time.

For the ice-shelf basal mass balance, we applied two melt-rate parametrisations $M_{b1}$ and $M_{b2}$, based on Gong et al. (2014) and Beckmann and Goosse (2003), respectively. The former is a scheme that allows the highest melt rates to follow the

grounding-line migration, using a combined function of ice thickness and distance to the grounding line, defined as

$$M_{b1} = H^\alpha (pG + (1-p)A), \tag{1}$$

where $H$ is the ice thickness, and $G$ and $A$ are tuning parameters to constrain melt rates at the grounding line, and away from the grounding line, respectively. The value of $p$ decreases exponentially with distance to the grounding line, taking the value

of 1 at the grounding line and 0 away from it (Cornford et al., 2015, Appendix B2), and $\alpha$ is a tuning parameter. The $M_{b2}$ parametrisation is based on the difference between the freezing point of water and ocean temperature near the continental shelf break (as developed in Beckmann and Goosse, 2003). The virtual temperature $T_f$ at which the ocean water freezes at the depth $z_b$ below the ice shelf is defined as

$$T_f = 273.15 + 0.0939 - 0.057S_o + 7.64 \times 10^{-4} z_b, \tag{2}$$

where $S_o$ is the ocean salinity (set at 34.5 psu from Schmidtko et al., 2014, confirmed by K. Leonard, personal communication, 2012). The melt rates $M_{b2}$ are prescribed as

$$M_{b2} = \frac{\rho_w c_{p_0} \gamma_T F_{melt}(T_0 - T_f)}{L_i \rho_i} \tag{3}$$

where $\rho_w$ is the density of water, $c_{p_0}$ the specific capacity of the ocean mixed layer, $\gamma_T$ the thermal exchange velocity, $T_0$ the ocean temperature (set at -1.5 °C from Schmidtko et al., 2014, and K. Leonard, personal communication, 2012), $L_i$ the latent

heat capacity of ice, $\rho_i$ the density of ice (Table 1 for the value of parameters) and $F_{melt}$ a tuning parameter.

Ice temperature data are provided by a three-dimensional thermo-mechanical model (updated from Pattyn, 2010) and are constant in time.

## 2.2   Ice-sheet modelling

The simulations were performed using the finite volume ice-sheet model BISICLES (http://BISICLES.lbl.gov). The model

solves the Shallow Shelf Approximation (SSA) and includes vertical shearing in the effective strain rate (the model is fully detailed in Cornford et al., 2015), which makes the ice softer than the traditional SSA approach at the grounding line, and induces similar ice sheet behaviour compared to full-Stokes models (Pattyn and Durand, 2013) when using sub-kilometric resolution at the grounding line. We assessed the sensitivity to the grid resolution at the grounding line, between 250 m and 4000 m (Supplementary Figure 2). The contribution to sea level change and grounding-line migration converge below 500 m.

We thus used 500 m resolution at the grounding line for all the transient simulations, up to 4000 m farther inland (Table 2.1). The equations are solved on an adaptive horizontal 2-D grid rendered by the Chombo framework. Data assimilation is performed by a control method that solves the adjoint system of equations, as described in Appendix B1 of Cornford et al. (2015). The relationship between stresses and strain rates is given by the Glen's flow law:

$$\boldsymbol{S} = 2\phi\eta\dot{\boldsymbol{\varepsilon}}, \tag{4}$$

**Table 1.** Model parameters

| Parameter | Symbol | Value | Unit |
|---|---|---|---|
| Ice density | $\rho_i$ | 917 | kg m$^{-3}$ |
| Water density | $\rho_w$ | 1028 | kg m$^{-3}$ |
| Gravitational acceleration | $g$ | 9.81 | m s$^{-2}$ |
| Glen's exponent | n | 3 | |
| Basal friction exponent | m | (1, 1/3) | |
| Grid resolution | | 4000 down to 500 | m |
| Specific heat capacity of ocean mixed layer | $c_{p0}$ | 3974 | J kg$^{-1}$ K$^{-1}$ |
| Thermal exchange velocity | $\gamma_T$ | $10^{-4}$ | m s$^{-1}$ |
| Temperature of the ocean | $T_0$ | 271.65 (-1.5 °C) | K |
| Salinity of the ocean | $S_O$ | 34.5 | psu |
| Latent heat capacity of ice | $L_i$ | 3.35 $10^5$ | J kg$^{-1}$ |
| Tuning parameter for $M_{b1}$ | $\alpha$ | 3 | |
| Tuning parameter for $M_{b1}$ | G | (25, 50, 100) $10^{-9}$ | |
| Tuning parameter for $M_{b1}$ | A | 0 | |
| Tuning parameter for $M_{b2}$ | $F_{melt}$ | (0.01, 0.02, 0.03) | |

where $S$ is the deviatoric stress tensor, $\dot{\varepsilon}$ is the strain rate tensor, $\eta$ is the effective viscosity (depending on englacial temperatures and effective strain rate), and $\phi$ is a stiffening factor representing non-thermal viscosity effects, such as crevasse-weakening and ice anisotropy. The basal friction between the grounded ice sheet and the bed is governed by a Weertman-type sliding law (Weertman, 1957):

$$5 \quad \boldsymbol{\tau}_b = \begin{cases} -C|\boldsymbol{u}_b|^{m-1}\boldsymbol{u}_b & \text{if} \quad \dfrac{\rho_i}{\rho_w}H > -b \\ 0 & \text{otherwise} \end{cases} \quad (5)$$

where $\boldsymbol{\tau}_b$ is the basal traction, $C$ is the friction coefficient, $m$ is the friction exponent and $\boldsymbol{u}_b$ is the basal velocity. Initial fields of $C$ and $\phi$ were both inferred (simultaneously and over the entire computational domain) with the control method applied to ice/bed geometry and surface velocities, using the same procedure as described in Berger et al. (2016), as well as the results obtained for the $C$ and $\phi$ fields.

## 2.3 Description of the experiments

**Initialisation**

Three sets of initialisations with both linear ($m = 1$) and nonlinear ($m = 1/3$) sliding were performed for $C$, $\phi$ (both inferred with the control method), and the initial ice/bed geometry:

- $B_e/S$: The control method and the transient simulations use the *high-resolution* dataset (PPw is included in model initialisation and evolution).

- $B_e/U$: This is a variant of $B_e/S$ in which transient simulations start from bed elevation and ice thickness without resolving PPw - we use Bedmap2 instead of mBedmap2 - in order to simulate unpinning.

- $RF/S$: The control method and the transient simulations use the *standard* dataset (PPw is excluded from initialisation and evolution).

Because there is no friction beneath ice shelves, we set the value of the friction coefficient $C$ in case of further ice-shelf re-grounding during transient simulations at $500$ Pa m$^{-1}$ $a$. This number causes high basal sliding (comparable to sliding beneath ice streams), which reflects the idea of a sediment-filled bathymetry, and is motivated by the sediment layer inferred from airborne radar and ice-sheet modelling upstream of the WRG grounding line (Callens et al., 2014).

After model initialisation, the ice-sheet geometry was relaxed for 50 years prior to the transient simulations in the same manner as the first stage of relaxation in Cornford et al. (2015), which was enough to decrease the ice-flux divergence due to artefacts of interpolation and other sources of geometry errors. To relax the ice sheet, we thus applied sub-ice shelf melt rates computed from mass conservation assuming steady state, which gives values in line with current observations (Depoorter et al., 2013; Rignot et al., 2013). However, applying such melt rates beneath the HB ice shelf leads to a rapid retreat of the grounding line induced by a MISI during the time span of the relaxation. We solved this issue by applying a positive basal mass balance (i.e., accretion) of $1$ m a$^{-1}$ during the relaxation, which helps to stabilise the ice shelf, but leads to few kilometres advance of the grounding line (stabilising the ice sheet during relaxation can also be done by fixing the ice shelf thickness, such as in Arthern et al., 2015, and Wright et al. (2014)). Surface elevation change rates (and their spatial gradients) drop by an order of magnitude (Figure 2) during relaxation.

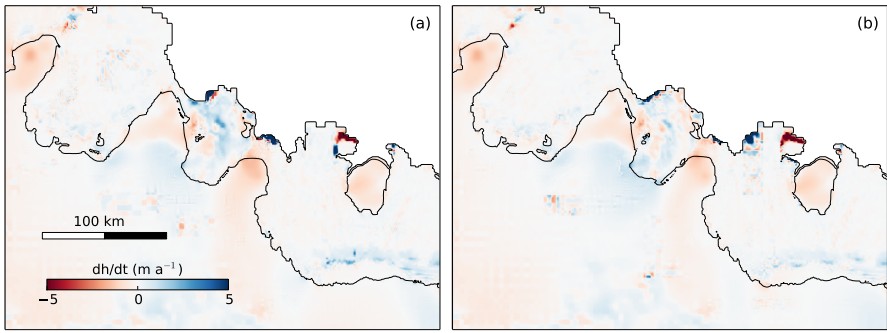

**Figure 2.** Surface elevation change rates after relaxation of $Be/S$ and $Be/U$ (a) and $RF/S$ (b) initialisations, for linear sliding.

**Table 2.** Setup of all 36 experiments. The name of each experiment reflects the dataset used for initialisation, its initial ice/bed geometry, the form of sliding law, and the type and amplitude of the melt-rates.

| Experiment name | Dataset for | | m | Sub-ice shelf melt rates | |
| --- | --- | --- | --- | --- | --- |
| | Data assimilation | Initial geometry | | Type | Amplitude |
| $B_e/S/L/M_{bi}/A_j$ | *high-resolution* | *high-resolution* | 1 | | |
| $B_e/S/NL/M_{bi}/A_j$ | | | 1/3 | | |
| $B_e/U/L/M_{bi}/A_j$ | *high-resolution* | *standard* | 1 | i=(1,2) for | j=(l,m,h) for |
| $B_e/U/NL/M_{bi}/A_j$ | | | 1/3 | $M_{b1}$ or $M_{b2}$ | (low, medium, high) |
| $RF/S/L/M_{bi}/A_j$ | *standard* | *standard* | 1 | | |
| $RF/S/NL/M_{bi}/A_j$ | | | 1/3 | | |

### Transient scenarios

Each initialisation is followed by 12 different transient simulations, applying either linear or nonlinear sliding together with 6 different prescribed sub-ice shelf melt rates, $M_{b1}$ and $M_{b2}$, each with 3 different amplitudes - low, medium and high - set by tuning the parameters $\alpha$, G and A for $M_{b1}$, and $F_{melt}$ for $M_{b2}$ (Table 1). The naming convention adopted for transient simulations and the corresponding parameters are given in Table 2.

The sum of initial medium melt rates over the ice shelves yields values that are comparable to current values (Rignot et al., 2013; Depoorter et al., 2013, and M. Depoorter, personal communication, 2016; Supplementary Table). The sum of low and high melt rates represent approximately half and twice the sum of medium melt rates, respectively. Initial melt rates $M_{b1}$ and $M_{b2}$ of medium amplitude are shown in Figure 3 for the $B_e/S$ initialisation. For similar amplitudes, $M_{b1}$ causes much higher melt rates than $M_{b2}$ close to the grounding line, where melt rates are always the highest whatever the type of melt rates.

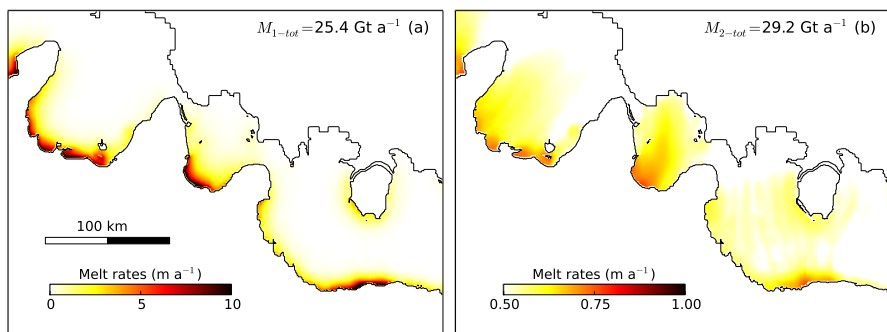

**Figure 3.** Initial fields of medium $M_{b1}$ (a) and $M_{b2}$ (b) sub-ice shelf melt rates for the $B_e/S$ initialisation. The sum of melt rates over the computational domain, written at the top right of panels, is comparable to current values (Rignot et al., 2013; Depoorter et al., 2013, and M. Depoorter, personal communication, 2016; Supplementary Table).

# 3 Results

## 3.1 Data assimilation

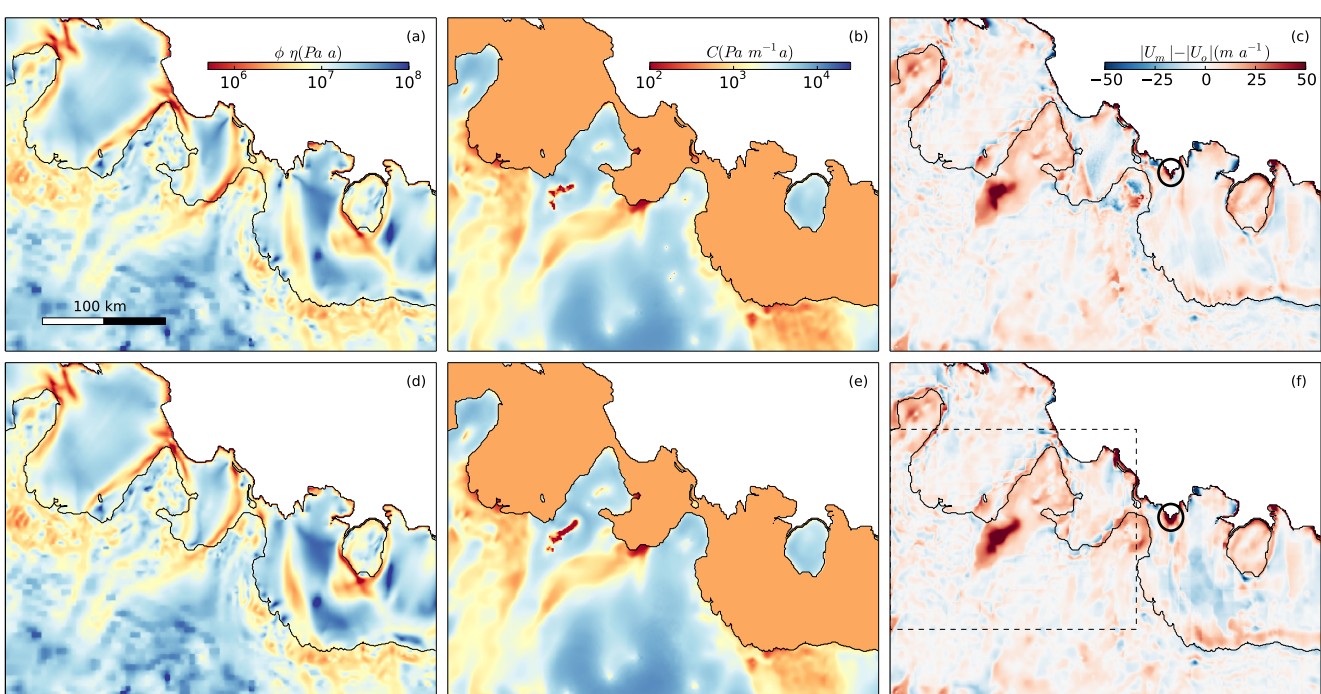

**Figure 4.** Results of the control method for $B/S$ and $B/U$ (a,b,c) and $RF/S$ (d,e,f) initialisations, for linear sliding. Vertically averaged effective viscosity (a,d), basal friction coefficient (b,e) (for current ice shelves, the value of $C = 500\ Pa\ m^{-1}\ a$ is prescribed for transient simulations) and difference between modelled and observed velocities (c,f). The circles indicate PPw (c,f). The dashed box (f) marks the large mismatch that are discussed in the text and shown in more details in Supplementary Figure 1.

The L-curve analysis performed by Berger et al. (2016) to optimise regularisation still holds for our extended domain and nonlinear sliding, even though it was originally applied to a smaller domain and linear sliding.

The root mean square error between modelled and observed velocities after data assimilation is $\approx 14$ m a$^{-1}$ for $B_e/S$ and $B_e/U$ initialisations, and $\approx 13$ m a$^{-1}$ for $RF/S$ initialisation, and is independent of the applied sliding law. Such mismatches are similar to what was already computed by control methods applied to the Antarctic ice sheet (e.g. Fürst et al., 2015; Cornford et al., 2015). The largest mismatches are found at the calving front, on ice rises and promontories, as well as upstream of the TB/HB promontory (Figure 4). We attribute the latter to the poor consistency between the high observed surface slope and thickness combined with low surface velocities (Supplementary Figure 1), as high driving stresses should induce high velocities. The control method cannot deal with such a non-physical combination for a steady-state ice sheet: it decreases the

friction during the first iterations, and further attempts to catch up with the consequent mismatch through ice stiffening during the following iterations.

A significant difference between the two datasets appears in the vicinity of PPw (Figure 4), where the mismatch is lower when using the *high-resolution* dataset. There, omitting PPw in the control method leads to an excessive ice stiffening (Figure 5
in Berger et al., 2016, and hereafter in Figure 7).

The central parts of ice shelves are comparatively more viscous, except within rifting areas, where the viscosity can be few orders of magnitudes smaller. The friction coefficient is comparatively small beneath the ice streams of WRG, HB and TB, and few orders of magnitude higher where ice velocity is small, such as in between ice streams and beneath ice promontories and rises. We show these results in Figure 4 with linear sliding.

## 3.2 Initial speed up after unpinning

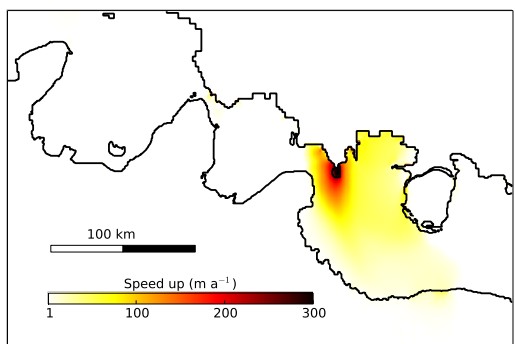

**Figure 5.** Speed up due to unpinning after 50 a for medium melt rates $M_{b1}$ and linear sliding. Absolute velocity differences (m a$^{-1}$) between $B_e/U/L/M_{b1}/A_m$ and $B_u/U/L/M_{b1}/A_m$.

Unpinning (for $B_e/U$ initialisation) induces an instantaneous acceleration of the WRG ice shelf by up to 300 m a$^{-1}$ at the former location of PPw. After 50 a, the acceleration has propagated over almost the entire ice shelf up to the grounding line, but none of the neighbouring ice shelves of HB and East Ragnhild Glacier is affected by unpinning (Figure 5). The central flowline of the WRG ice stream migrates westward and relocates at an almost equal distance from the HB/WRG promontory
and DIR within a few years. The velocities at the ice-shelf front are $\approx$ 20% larger than for $B_e/S$ initialisation. Overall, the comparatively faster ice shelf induces a less advanced grounding line at the end of simulations (about 10 km). The velocity increase near the HB/WRG promontory leads to thinning of its eastward side, making its saddle area afloat and turning it into an ice rise more rapidly than for $B_e/S$ and $RF/S$ initialisations.

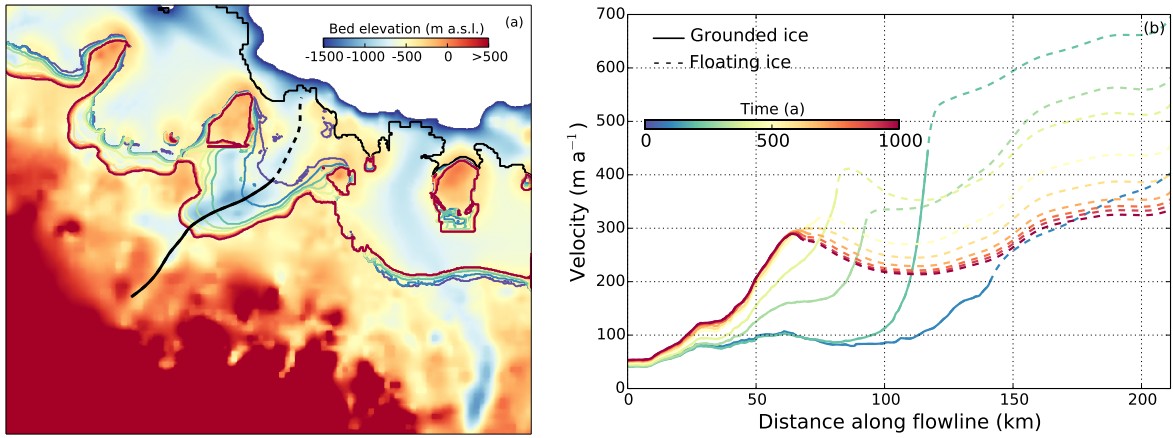

**Figure 6.** Grounding-line migration for the $B_e/S/L/M_{b1}/A_m$ experiment. (a) Bed elevation in the background, grounding lines are shown every 100 years (colorscale shown in (b)) and the dashed line shows the central flowline of HB. (b) Ice velocity profiles along the central flowline of HB, shown every 100 years. The grounding-line position is marked by the limit between solid and dashed parts of profiles.

### 3.3  Main steps of grounding-line migration

The grounding line migrates similarly for medium melt-rates experiments with linear sliding (Figure 7) and nonlinear sliding. Here we present the common successive steps of all scenarios regarding grounding-line migration and ice dynamics (Figure 6 and Supplementary Movie).

The HB ice shelf/sheet system is by far the most dynamic of the three glaciers. During the first century, its grounding line retreats relatively slowly and the pinning points PPhs (Figure 1) detach from the ice-shelf base. The subsequent unpinning of PPhs is followed by an acceleration of the grounding-line retreat over the deepest part of the bed, along with a speed up of ice increasing from $\approx 20\%$ to 100% in a hundred year or so. During these rapid changes, two sudden jumps (the second being less strong than the first) in velocity and grounding-line retreat rates occur when the grounding line retreats over two consecutive

troughs imprinting the bed. During the following years, the grounding line and velocities of HB stabilise progressively as the grounding line gets closer to the downsloping part of the bed. By the end of the simulations, the two saddles linking the TB/HB and HB/WRG promontories to the main ice sheet get successively afloat until the two promontories transition into ice rises, and the grounding line of HB has retreated by up to 100 km. The consequent loss of buttressing eventually produces a small retreat of the TB grounding line for the highest melt rates scenarios.

The $B_e/U$ initialisation produces faster retreat of grounding lines than the $B_e/S$ initialisation, which produces faster retreat than the $RF/S$ initialisation. In particular, the saddle of the HB/WRG promontory gets afloat the most rapidly. The grounding lines of TB and WRG re-advance over up to tens of kilometres for low-melt scenarios.

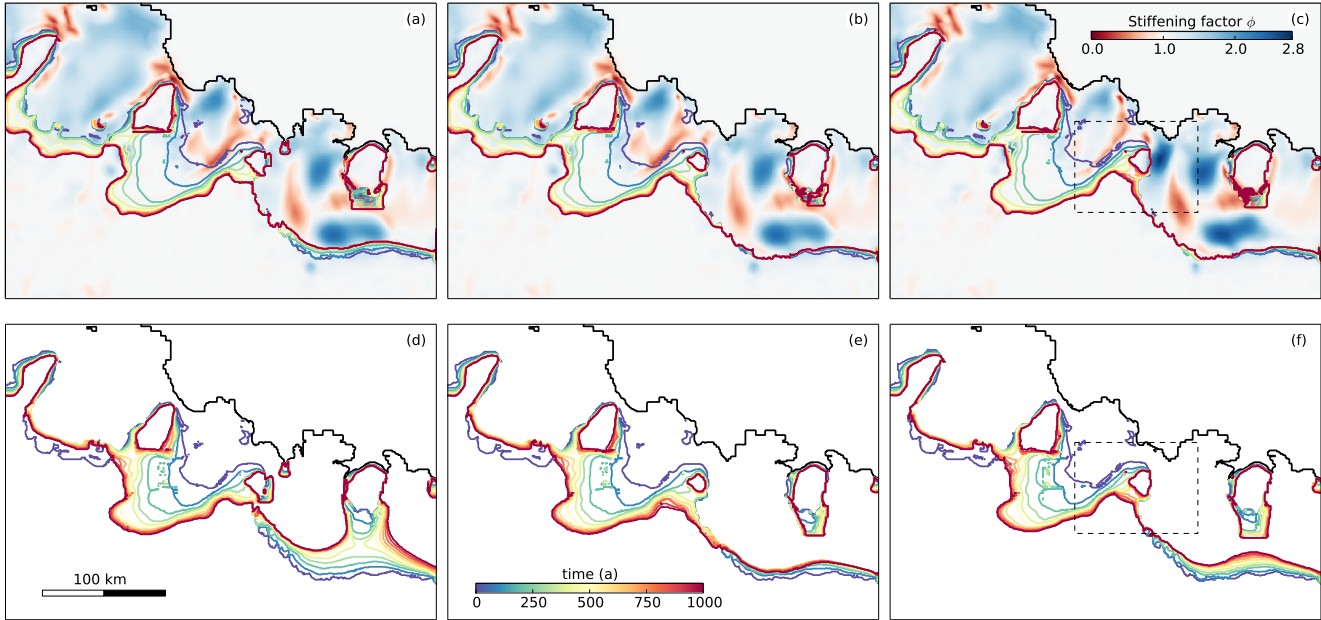

**Figure 7.** Grounding-line migration for medium melt rates and linear sliding. Melt rates $M_{b1}$ (a,b,c) and $M_{b2}$ (d,e,f). Experiments $B_e/S/L/M_{b1}/A_m$ (a), $B_e/U/L/M_{b1}/A_m$ (b), $RF/S/L/M_{b1}/A_m$ (c), $B_e/S/L/M_{b2}/A_m$ (d), $B_e/U/L/M_{b2}/A_m$ (e) and $RF/S/L/M_{b2}/A_m$ (f). Grounding lines are shown every 100 years. In (a,b,c) the stiffening factor field is shown in the background and a dashed line window is drawn to point out the area where excessive stiffening occurs when omitting PPw in data assimilation.

## 4  Discussion

Most of the continental shelf beneath the WAIS lies below sea level, making the ice sheet prone to undergo a MISI (Ritz et al., 2015; Mercer, 1978; Joughin and Alley, 2011). With respect to the bed topography, the EAIS appears more stable, but its volume of ice is ten times larger than its western counterpart. It is therefore crucial to investigate a potential unstable
5   retreat of grounding lines that may further affect the ice-sheet stability. Here, our simulations systematically show an unstable retreat of HB over the first few hundreds years regardless of the applied sub-ice shelf melt rates, sliding laws and initialisations (Figure 7 and Supplementary Movie). Half of the simulations also predict the retreat of the neighbouring glacier TB for melt rates comparable to current observations. Overall, the contribution of the studied area to sea-level rise is $30 \pm 10$ mm for the next millennium, which needs to be put in perspective with the comparatively small domain (representing about 1% of the
10   Antarctic ice sheet) and the possible nonlinear effects due to future oceanic forcing that are neglected in this study.

After a century, the HB grounding-line retreat reaches its highest speed, 1 km a$^{-1}$ for the $B_e/S/L/M_{b1}/A_m$ experiment and 500 m a$^{-1}$ for the $B_e/S/L/M_{b2}/A_m$ experiment, further inducing after a couple of decades a peak in ice-shelf velocities, attaining 700 m a$^{-1}$ for the latter two experiments when the grounding line retreats over the deepest part of the bed (Figure 6). The retreat is thus influenced by the type and amplitude of melt rates (Figure 8). We also evaluated the MISI part on the

retreat of HB, by switching off the sub ice-shelf melting during the $B_e/S/L/M_{b1}/A_m$ when the grounding line retreats over the upsloping part of the bed, without altering the melt rates beneath the other ice shelves. The experiment, shown in Supplementary Figure 3, demonstrates that the grounding line retreat is substantially affected by a MISI, even though not entirely. However, none of the simulations show a retreat of the WRG grounding line, despite the presence of an incised valley of about 1200 m deep beneath the ice upstream of the grounding line (Figure 1). This valley is also narrow and starts tens of kilometres upstream of the current grounding line, while the depression beneath the HB grounded ice is wider and starts closer to the grounding line. This accords with the ideal simulations of Gudmundsson et al. (2012), who showed that a wider trough upstream of a grounding line reduces the buttressing exerted by the ice shelf, which enhances the grounding-line retreat rate.

During the retreat of HB, the ice-shelf thickness is halved compared to initial conditions. Meanwhile, the thickness of the WRG ice shelf remains almost constant in time near the east side of the HB/WRG promontory. The consequence is an increase of the ice flux coming from the promontory's saddle and going towards the HB ice shelf, reducing the width of the saddle from its western side and eventually transiting the HB/WRG promontory into an ice rise when its saddle becomes afloat. The retreat of TB depends on the melt-rates type and amplitude. All the low amplitude and the $M_{b2}$ medium amplitude melt rates lead to an advance of its grounding line, while the other scenarios lead to a retreat. However, this contrasting behaviour only slightly modulates the time span by which the saddle of the TB/HB promontory gets afloat, for which the substantial thinning of the HB ice shelf is the major driver.

The observed grounding lines that are currently fringed and buttressed by ice promontories (such as for HB) are stable in the studied area, even resting on upsloping bed (also shown by Gudmundsson et al., 2012, for synthetic numerical experiments). However, small amounts of sub-ice shelf melting clearly induce rapid grounding-line retreat and transition of the promontories into ice rises. Such a quick transition is corroborated by Favier and Pattyn (2015), showing that promontories are transient features of grounding-line retreat, when they are characterized by an overdeepening upstream of the pinned area.

Most low and several medium melt rates scenarios lead to an advance of the WRG grounding line upstream of DIR (Figure 7), even though we excavated the area beneath the ice shelf. Because the bathymetry of ice-shelf cavities is poorly constrained, advancing grounding lines must be cautiously interpreted, and this potentially spurious effect on sea level thus calls for a better knowledge of bathymetry beneath ice shelves.

Unpinning of the WRG ice shelf mildly affects the global contribution to sea level, increasing it by 10% compared to the $B_e/S$ initialisations (Figure 8). However, the decrease of buttressing stemming from unpinning thins the WRG ice shelf and accelerates the retreat of the HB/WRG promontory's saddle from its eastern side: the saddle gets afloat two centuries earlier (Figure 8). This indicates a large sensitivity of promontories deglaciation to a loss of buttressing, similarly to the unstable retreat pointed out in Favier and Pattyn (2015). The loss of buttressing induced by unpinning also cancels the advance of the WRG grounding line simulated by the experiments using $B_e/S$ initialisations (Figure 7b), but does not have enough effect to induce an unstable retreat over the upsloping bed area upstream of the grounding line. On the west side of the HB/WRG promontory, unpinning of PPhs occurring after less than 100 years of simulation precedes by few years the acceleration of the ongoing retreat of the HB grounding line (Supplementary Movie). However, quantifying the contribution of PPhs unpinning

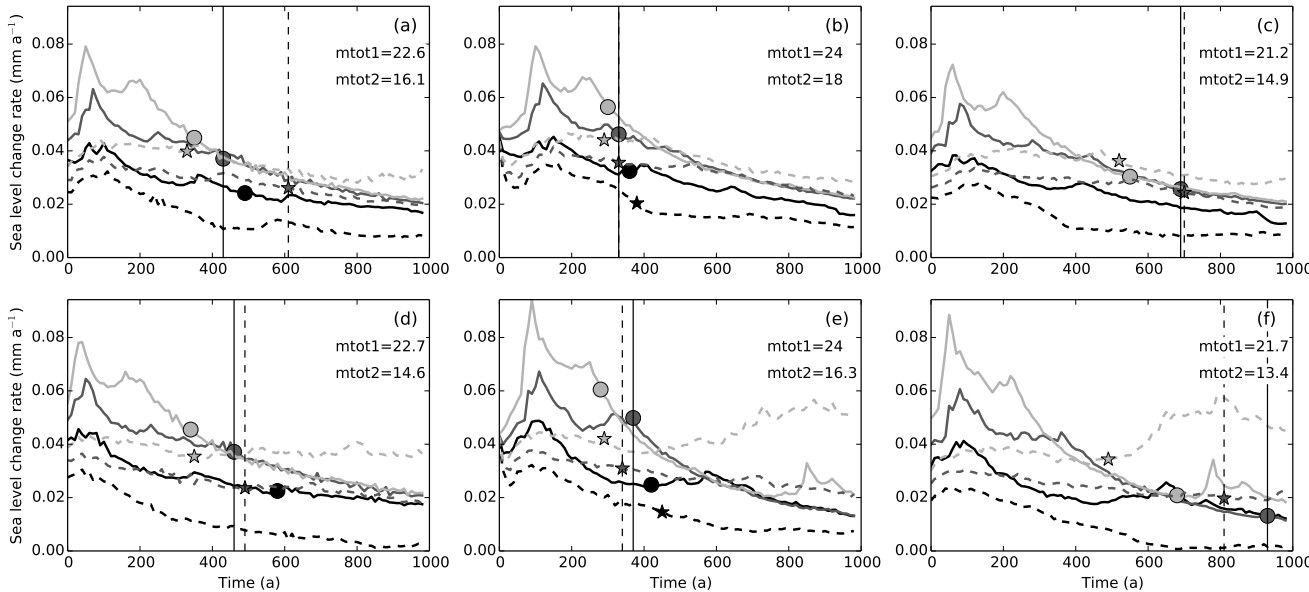

**Figure 8.** Contribution to sea level for all transient simulations. Linear sliding (a,b,c) and nonlinear sliding (d,e,f) experiments. Experiments using $B_e/S$ (a,d), $B_e/U$ (b,e) and $RF/S$ (c,f) initialisations. Solid and dashed lines show $M_{b1}$ and $M_{b2}$ melt rates, respectively. The brighter the line, the higher the melt-rate amplitude. The circles (in $M_{b1}$ lines) and triangles (in $M_{b2}$ lines) markers indicate the time by which the HB/WRG promontory transitions into an ice rise, which is also marked by a vertical line (solid or dashed) for the medium melt rates experiments. The two numbers shown at the top right of each panel indicate the contribution to sea-level change in mm after 500 a of medium melt rates experiments ($mtot1$ for $M_{b1}$, and $mtot2$ for $M_{b1}$).

to the grounding-line retreat is not straightforward, since unpinning is effective when the HB grounding line retreats over the deepest, hence with the largest potential for inducing a MISI.

Besides the MISI-driven consequences on sea level, sub-ice shelf melting is the other main driver of the retreat. Different behaviours emerge from the two types of melt-rate parametrisations. During the first few hundreds of years, sea-level contri-
5  bution is more or less a linear function of melt-rates amplitude. The form of $M_{b1}$ induces high melt rates at the grounding line when it retreats over the deep trough beneath HB. The contribution to sea level is then a function of pure melting and dynamic thinning, inducing peaks of sea-level contribution after about a century. In the case of $M_{b2}$ melt rates, this peak is replaced by a milder bump in sea-level contribution (Figure 7) since the pure-melting contribution is lower. After 500 a, the retreat of the HB grounding line is less rapid and the contribution to sea level is then mostly due to melting, and to a lesser extent due to dynamic
10  thinning. Since the $M_{b1}$ melt rates induce more melting at large depth and almost no melting closer to the surface compared to the $M_{b2}$ of similar amplitudes, the $M_{b1}$ melt rates become lower compared to the $M_{b2}$ melt rates, except for the lowest melt rates where this is the opposite. After $\approx 800$ a, a sudden increase in sea-level contribution occurs for nonlinear sliding and the high $M_{b2}$ melt rates (Figure 8(e,f)), which is due to ungrounding of the promontory saddle between the Lazarev ice shelf and

the Unnamed Glacier (Figure 1). This peculiar behaviour that does not occur for the other experiments is the reason why we indicate the sea-level contribution after 500 a of transient simulation (Figure 8).

Compared to linear sliding, nonlinear sliding (with $m = 1/3$) should enhance basal sliding when ice velocity increases. The acceleration of HB during its retreat consequently yields higher velocities and faster retreat rates of the grounding line for the nonlinear case, hence leading to a higher contribution to sea level from HB. At the scale of the domain, this is however difficult to distinguish the differences between the two sliding laws, since they induce similar contributions after 500 a.

As already shown by Berger et al. (2016), omitting the pinning point PPw in data assimilation induces erroneous ice stiffening nearby (Figure 7). Initialising transient simulations with such stiffening leads to a spurious decrease in sea-level contribution by 10% compared to the experiments using $B_e/S$ initialisation. The transient evolution of the WRG grounding line looks similar to the unpinning experiments, pointing out the spatially limited effects of the excessively stiffened ice. However, the stiffening effect largely alters the timing of deglaciation of the HB/WRG promontory (Figure 8) and delays it by approximately 500 a. Moreover, any further local change in the boundary condition between the pinning points and the ice shelf, including the extreme - but possible - event of unpinning (for instance induced by a substantial thinning of ice shelves Paolo et al., 2015) cannot be simulated by the model if the pinning point is omitted in the first place.

Since the early 2000s, uncertainties of ice-sheet modelling outputs have been reduced by substantial numerical improvements, enabling to grasp more accurately key processes such as grounding-line migration (Pattyn and Durand, 2013). This improvement was also made possible by the increasing computational power. We are now able to simulate the behaviour of the WAIS using higher order models at a high spatial resolution in the relevant areas for a wide range of scenarios over the next centuries (Cornford et al., 2015), which was not feasible a few years ago. Nevertheless, the lack of knowledge of essential parameters still affects simulations of the Antarctic ice sheet behaviour, hence preventing further decrease of uncertainties in sea-level predictions. Sub-ice shelf melting is a major driver of ice-sheet retreat and sea-level contribution (Figure 8). Even though forcing the ice sheet with parametrised melt rates (such as in this study) gives qualitative and informative insights on future sea-level contribution, the lack of knowledge of the cavity beneath ice shelves prevents the use of more advanced assessment based on ocean modelling (such as in Hattermann et al., 2014; De Rydt and Gudmundsson, 2016). Moreover, the ill-constrained shape of the ice-shelf cavity dictates how and if the grounding line advances, which also biases future sea-level predictions. Here, we demonstrates that sea-level predictions and timing of deglaciation can be substantially affected by a too shallow bathymetry and the absence of small pinning points, which all affect ice-sheet initialisation. Also, the exact representation of pinning points (ice rumples, rises and promontories) in the observational datasets, even if they are small, is key for more accurate predictions of future sea-level change and timing of ice-sheet retreat. Therefore, improving these predictions by the use of ice-sheet modelling relies on future improvements of our knowledge of the bathymetry beneath ice shelves and (small) pinning points.

## 5    Conclusions

We use the ice-sheet model BISICLES to evaluate the contribution of the outlet glaciers between the Lazarev and Roi Baudouin ice shelves in East Antarctica to future sea-level rise, with two different input datasets including or excluding an observed small pinning point (PPw) at the calving front. We also investigate the influence of various sub ice-shelf melt rates parametrisation and two types of Weertman-like sliding law (linear and nonlinear). Our results suggest an unstable retreat of the Hansenbreen (HB) glacier within the next century. This retreat is equally driven by sub ice-shelf melting and marine ice sheet instability (MISI), while the other outlet glaciers are relatively stable over the next millennium. Where the bed is downsloping towards the sea (no potential for a MISI), sub-ice shelf melting exclusively controls sea-level contribution. Unpinning of PPw increases the sea-level contribution by 10% but substantially affects the timing of ice-sheet retreat in the most sensitive parts, such as the HB/WRG promontory which transitions into an ice rise 200 a in advance. On the other hand, omitting PPw during the initialisation of the ice sheet yields local excessive ice-shelf stiffening, which decreases the sea-level contribution by 10% and delays the HB/WRG promontory transition by 500 a in transient simulations. Even small pinning points should be accounted for in ice-sheet modelling because they affect transient ice-dynamical behaviour and grounding-line retreat. This study calls for a better knowledge of Antarctic ice sheet margins, including the bathymetry beneath ice shelves and the characteristics - ice velocity and ice/bed geometry - of even the smallest pinning points, in order to improve our ability to predict future Antarctic ice sheet margins.

*Acknowledgements.*    This paper forms a contribution to the Belgian Research Programme on the Antarctic (Belgian Federal Science Policy Office), Project SD/CA/06A (Constraining Ice Mass Change in Antarctica, IceCon). R. Drews was partially supported by the Deutsche Forschungsgemeinschaft (DFG) in the framework of the priority programme "Antarctic Research with comparative investigations in Arctic ice areas" by the grant MA 3347/10-1. BISICLES development is led by D.F. Martin at Lawrence Berkeley National Laboratory, California, USA, and S.L. Cornford at the University of Bristol, UK, with financial support provided by the U.S. Department of Energy and the UK Natural Environment Research Council. Simulations were carried out using the computational facilities of the HPC computing center at Université libre de Bruxelles. We are grateful to M. Depoorter for providing us with current sub-ice shelf melt rates. We thank Robert J. Arthern and an anonymous reviewer for their insightful comments that helped to improve the manuscript.

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
