# Peer review of "Dynamic influence of pinning points on marine ice-sheet stability: *a numerical study in Dronning Maud Land, East Antarctica"

_The Cryosphere, 2016_

## Referee Comment (RC1) · R. Arthern (Referee) · 29 Jul 2016

R. Arthern (Referee)

rart@bas.ac.uk

This study describes glaciological simulations of a region of coastal East Antarctica. The simulations are performed using the BISICLES ice sheet model that has previously been shown to resolve stresses and velocities accurately at the grounding line when run at sufficiently high resolution. A novel feature of this study is the investigation of several different melt rate parameterisations and an investigation into the consequences of pinning points where detailed local surveys show that the ice is grounded, but the available continental scale bathymetric charts would suggest otherwise. The paper is logically organised and clearly written. The sensitivity studies performed make sense, the figures are appropriate and the conclusions are important enough to be

published in The Cryosphere.

My main concern with this paper is that I think more needs to be done to demonstrate that the results converge under grid refinement. The effect of grid resolution might be especially important if basal melting makes a sudden jump across the grounding line from a finite value to zero, or if the basal melt depends sensitively on ice draft that varies rapidly near the grounding line. Both of these conditions are relevant here and the melt parameterisations are different from those used previously, so I don't think it is enough to rely on previous investigations with this model to assess the sensitivity to grid refinement.

Major considerations

There are at least three motivations for investigating the sensitivity to grid resolution more thoroughly than is done here.

1 km resolution at the grounding line seems quite coarse unless a sub-grid scheme is used to parameterise basal drag and driving stress at the grounding line. Was a sub-grid parameterisation used? Are the results sensitive to the mesh refinement?

Another important consideration is that both melt rate parameterisations have the potential to apply non-zero melt rate directly at the grounding line, with a sudden transition to no melt for grounded ice. This might impose a discontinuity in the gradient of ice thickness at the grounding line. Again, it would be good to see evidence that the model can resolve this adequately.

Also, both parameterisations depend on ice draft. Basal slopes at the grounding line can be steep. Again, this means the authors should show that their results do not depend sensitively on the grid refinement used.

I think a simulation should be included with a modified parameterisation for which the melt rate goes to zero at the grounding line (perhaps by setting G=0 in Equation 1). This would reveal whether the retreat is driven mostly by melting directly at the grounding
line, or by reductions in buttressing induced by melting elsewhere.

Minor corrections

P1 - Line 22. This section includes the following statement

A prior retreat of the grounding line (e.g. ocean driven) resting on such an upsloping bed thickens the ice at the grounding line, which increases the ice flux and induces further retreat, etc., until a downsloping bed is reached.

This description is misleading. The retreat needn't stop when a downsloping bed is reached. In the simulations of Schoof (2007) the grounding line doesn't stop at the deepest point, which is where a downsloping bed is first encountered. There is the possibility for a stable equilibrium to exist on downsloping beds, but only if upstream snowfall balances local flux at the grounding line. There is no reason for this condition to apply just because the grounding line has reached a downsloping bed. Rather, the grounding line will continue to retreat until (i) upstream snowfall balances local flux at the grounding line AND (ii) the bed is downsloping. Some glaciologists seem to think that grounding line retreat will necessarily stop when a downsloping bed is reached, modellers shouldn't be adding to this confusion.

P2 – Line 25. Make clear compression/extension is in flow direction.

P5 – Explain more clearly which domain is bounded by the dotted line.

P5 – Eqn 1. I don't think G and A are melt rates. What values were used for these parameters?

P6 – How does p vary? Give more details of exponential decay rate.

P7 - Line 3. It needs to be clearer which equations are being solved. The text is currently ambiguous and leaves open three different possibilities. Is it (a) the L1L2 system described by Hindmarsh (2004), (b) the L1L2 system described by Schoof and Hindmarsh (2010) or (c) the SSA* system described by Cornford et al. (2015). If the

equations are the SSA* approach described by Cornford et al (2015) then these are not the same as the model described by Schoof and Hindmarsh (2010) and shouldn't be referred to as such.

P7 - Eqn 5. Is any regularisation used in the sliding law for low velocities?

P8 – Imposing a trial and error value of basal freezing to prevent grounding line motion during relaxation seems slightly imprecise. An alternative approach is to fix the thickness of floating ice shelf to prevent grounding line migration during relaxation (see Arthern et al. 2015, DOI: 10.1002/2014JF003239). This approach can also incorporate dhdt observations on the grounded ice so the grounding line retreats at the observed rate during the forward simulation. It would be worth pointing out that alternative approaches to constraining the grounding line during surface relaxation are available.

P16 – Line 9. Capitalise Weertman.

Table 1. Check the units for heat capacity.

Table 1. The parameter $\alpha_2$ is described as a tuning parameter for $M_{b2}$ not $M_{b1}$.

---

## Referee Comment (RC2) · Anonymous Referee #2 · 19 Aug 2016

**1  General statement**

The manuscript "Dynamic influence of pinning points on marine ice-sheets stability: a numerical study of Dronning Maud Land, East Antarctica" studies the impact of the presence of pinning points on outlet glaciers stability and their contribution to sea level rise. It shows that pinning points provide additional buttressing and therefore decrease the ice discharge by about 10%, but also that their presence strongly affects the ice shelf rheology inferred from data assimilation and therefore the model initial conditions. The authors suggest that including or omitting these pinning points impacts grounding line retreat and collapse of promontory on the timescale of several hundreds of years.

The manuscript is well written and the figures appropriate. The main point missing in the paper is that the authors do not show that their results are not resolution dependent. This can be easily done by rerunning a couple experiments with a higher mesh resolution and must be done in order to be confident that the results presented in this paper are robust. The interpretation of the impact of the melting is also quite ambiguous, as it differs in the discussion and conclusions. The section below describes in more details these two points and a few other specific comments.

**2 Specific comments**

p.1 l.16: "collapse" does not seem to be an appropriate word to describe a rather natural phenomenon that happens during simple grounding line retreat. This is also quite different from what is presented in the results.

p.2 l.11: The statement of ice rises not being detected by satellite observations is surprising: measurements of velocity and grounding line have a resolution of a few hundred meters while observations from altimetry also have an along track resolution a few hundred meters and the tracks are spaced by a few kilometers. Only bedrock topography does not have the required resolution, but sounding radars are operated on airborne and not satellite. So satellite observations, and in particular grounding line mapping should have the capability to resolve small ice rises and pinning points.

p.2 l.34: The L-curve analysis is described in more details in Jay-Allemand et al. (2011) and not Gillet-Chaulet et al. (2012).

p.3 l.13: The eastern ice shelf of Thwaites Glacier experienced a complex behavior during the past coupled decades, with successive periods of acceleration and deceleration (Mouginot et al., 2014) that are not coherent with the gradual grounding line retreat and unpinning of the eastern shelf ice rise (Rignot et al., 2014). Entrainment of the Eastern ice shelf by the main ice shelf and changes in the region between the

two parts of the ice shelf in this zone of intense shear is the preferred scenario to explain the complex changes observed and not a simple acceleration of this ice shelf (Mouginot et al., 2014).

p.4 l.25-30: How sensitive are the results to these two additional excavations?

p.7 l.7: Why limit the resolution to 1km? BISICLES does not have problems using improved resolution and the domain simulated is small enough that increasing the resolution to 500 or 250 m should not too much or a problem. This is especially surprising at this manuscript focuses on the impact of small ice rises and pinning points. Authors would need to show that their results are not resolution dependent by performing a couple of the simulations that experience large changes with a grid resolution divided by two (500 m or less at the grounding line).

p.7 l.16: How are the inversions of C and $\phi$ performed? Are they done simultaneously or one after the other? Are they done over the same region of different parts of the domain? This is not clear form the text and is an important question as changes in friction and stiffening factor can have a similar impact on ice flow.

p.8. l.10: How large is the melt rate in this region? It is surprising to see that adding just 1 m/yr drastically change the grounding line evolution from rapid retreat to small advance. This suggests that the model is very sensitive to this parameter.

p.12 l.17-18: This statement contradicts what is shown on Fig.7. The type of melt rate applied seems to play a significant role in at least the rate of grounding line retreat as the three upper plots with melting type $M_{b1}$ all have a similar grounding line evolution, while the three lower ones with melting type $M_{b2}$ also have a similar behavior, distinct from the previous one. And this is actually quite different from what is summarized in the conclusions.

**3 Technical comments**

p.2 l.6: "to more investigated" → "to be more investigated"

p.2 l.32: "overmatching" → "over fitting"

p.3 l.30: Which field measurements?

p.4 l.2: How do the first decades of the simulations compare with the observations that we have of the past couple decades?

p.4 l.3: How were these melt rates chosen? How do they compare with observations?

p.4 l.32: Authors should quickly summarize the model or observations that were used to derive this surface mass balance, and the year that is reproduces.

Fig. 2 caption: Notations should be consistent: B/S or $B_e$/S

p.7 l.24-25: Not clear. Simply say that you use the improved velocity and the standard bedrock topography maps.

p.8 l.19: What are the values from Rignot et al. (2013) and Depoorter et al. (2013) and what are the values used in the simulation for the different ice shelves? A table with the melt observed and used for each ice shelf could help make this comparison.

p.9 l.8: "mismatch" → "difference"

p.9 Fig. 3: This figure would be much clearer with colors.

p.11 Fig.5: Same as Fig.3, would be better with colors.

p.11 Fig.5 caption: "Velocity absolute" → "Absolute velocity"

p.12 l.7: There are many other very relevant references for the collapse of the WAIS.

p.12 l.13: "In total" → "Overall"

p.16 l.9: Weertman, capital missing

**4   References**

Depoorter, M. A., J. L. Bamber, J. A. Griggs, J. T. M. Lenaerts, S. R. M. Ligtenberg, M. R. van den Broeke, and G. Moholdt, Calving fluxes and basal melt rates of Antarctic ice shelves, Nature, doi:10.1038/nature12567, 2013.

Gillet-Chaulet, F., et al., Greenland ice sheet contribution to sea-level rise from a new-generation ice-sheet model, Cryosphere, doi:10.5194/tc-6-1561-2012, 2012.

Jay-Allemand, M., F. Gillet-Chaulet, O. Gagliardini, and M. Nodet, Investigating changes in basal conditions of Variegated Glacier prior to and during its 1982-1983 surge., Cryosphere, doi:10.5194/tc-5-659-2011, 2011.

Mouginot, J., E. Rignot, and B. Scheuchl, Sustained increase in ice discharge from the Amundsen Sea Embayment, West Antarctica, from 1973 to 2013, Geophys. Res. Lett., doi:10.1002/2013GL059069, 2014.

Rignot, E., S. Jacobs, J. Mouginot, and B. Scheuchl, Ice shelf melting around Antarctica, Science, doi:10.1126/science.1235798, 2013.

Rignot, E., J. Mouginot, M. Morlighem, H. Seroussi, and B. Scheuchl, Widespread, rapid grounding line retreat of Pine Island, Thwaites, Smith and Kohler glaciers, West Antarctica from 1992 to 2011, Geophys. Res. Lett., doi: 10.1002/2014GL060140, 2014.
* * *

---

## Author Comment (AC1) · 23 Sep 2016

**Response to the reviewers on manuscript tc-2016-144 - "Dynamic influence of pinning points on marine ice-sheet stability:**
a numerical study in Dronning Maud Land, East Antarctica" by L. Favier et al.

September 23, 2016

Lionel Favier, lionel.favier@ulb.ac.be

This document starts with the responses to the two reviewers on the manuscript tc-2016-144, then shows the corrections made on the paper and finishes with some Supplementary Material containing the new Supplementary Figures 2, 3 and the a new Supplementary Table.

**Response to Rob Arthern, Referee 1**

This study describes glaciological simulations of a region of coastal East Antarctica. The simulations are performed using the BISICLES ice sheet model that has previously been shown to resolve stresses and velocities accurately at the grounding line when run at sufficiently high resolution. A novel feature of this study is the investigation of several different melt rate parameterisations and an investigation into the consequences of pinning points where detailed local surveys show that the ice is grounded, but the available continental scale bathymetric charts would suggest otherwise. The paper is logically organised and clearly written. The sensitivity studies performed make sense, the figures are appropriate and the conclusions are important enough to be published in The Cryosphere.
We thank Robert Arthern for this very positive comment.

My main concern with this paper is that I think more needs to be done to demonstrate that the results converge under grid refinement. The effect of grid resolution might be especially important if basal melting makes a sudden jump across the grounding line from a finite value to zero, or if the basal melt depends sensitively on ice draft that varies rapidly near the grounding line. Both of these conditions are relevant here and the melt parameterisations are different from those used previously, so I don't think it is enough to rely on previous investigations with this model to assess the sensitivity to grid refinement.
You are right in your analysis. We therefore performed a sensitivity analysis on a grid resolution between 250 m and 4000 m at the grounding line. In terms of contribution to sea level rise, the 250 m and 500 m resolution give close results, while the 500 and 1000 m resolution are further away. We thus chose to present the 500 m resolution in the paper, and show the sensitivity results in new supplementary material (Supplementary Figure 2) showing sea level contribution and grounding line migration.

Major considerations

There are at least three motivations for investigating the sensitivity to grid resolution more thoroughly than is done here. 1 km resolution at the grounding line seems quite coarse unless a sub-grid scheme is used to parameterise basal drag and driving stress at the grounding line. Was a sub-grid parameterisation used? Are the results sensitive to the mesh refinement?

Another important consideration is that both melt rate parameterisations have the potential to apply non-zero melt rate directly at the grounding line, with a sudden transition to no melt for grounded ice. This might impose a discontinuity in the gradient of ice thickness at the grounding line. Again, it would be good to see evidence that the model can resolve this adequately.

Also, both parameterisations depend on ice draft. Basal slopes at the grounding line can be steep. Again, this means the authors should show that their results do not depend sensitively on the grid refinement used. No sub-grid parameterisation was used. See the previous comment for the sensitivity to resolution at the grounding line.

I think a simulation should be included with a modified parametrization for which the melt rate goes to zero at the grounding line (perhaps by setting G=0 in Equation 1). This would reveal whether the retreat is driven mostly by melting directly at the grounding line, or by reductions in buttressing induced by melting elsewhere. We use a parameterization of sub-ice shelf melt rates according to what is generally found in the literature, based on observations and/or ocean modelling. While the experiment with zero melt at the grounding line seems at first glance interesting, its parameterization is less straightforward in combination with established parameterizations. It would furthermore require a much larger sensitivity study on the melt rates to enable meaningful conclusions. Moreover, even though we can obtain the same amount of melting at the drainage basin scale for the two melting and no-melting at the grounding line equivalent scenarios, there may be two different evolutions for the ice draft, which would differentiate the two melting over time and complexify the interpretation. Actually, the best solution would be to use an ocean model to produce melt rates (we mention it now in the text and refer to the recent publication of De Rydt and Gudmundsson (2016)). However, those two solutions (ocean modelling and more parameterization) are clearly out of the scope of our paper. Given that such inclusion would change the scope of the paper considerably, we refrained from doing so. However, to answer the second Reviewer about the effect of the different melt rates types, we decided to run a simulation for which we switch off the sub ice shelf melting once the grounding line has started to retreat over the retrograde slope area of the Hansen glacier (melting is not altered beneath the other ice shelves). The results are now in a new supplementary figure which shows that the grounding line keeps retreating, even though not as fast as with melting. This somehow partially answers your question.

Minor corrections

P1 - Line 22. This section includes the following statement A prior retreat of the grounding line (e.g. ocean driven) resting on such an upsloping bed thickens the ice at the grounding line, which increases the ice flux and induces further retreat, etc., until a downsloping bed is reached. This description is misleading. The retreat needn't stop when a downsloping bed is reached. In the simulations of Schoof (2007) the grounding line doesn't stop at the deepest point, which is where a downsloping bed is first encountered. There is the possibility for a stable equilibrium to exist on downsloping beds, but only if upstream snowfall balances local flux at the grounding line. There is no reason for this condition to apply just because the grounding line has reached a downsloping bed. Rather, the grounding line will continue to retreat until (i) upstream snowfall balances local flux at the grounding line AND (ii) the bed is downsloping. Some glaciologists seem to think that grounding line retreat will necessarily stop when a downsloping bed is reached, modellers shouldn't be adding to this confusion. The reviewer is right, the sentence has been rewritten as advised.

P2 – Line 25. Make clear compression/extension is in flow direction. Done.

P5 – Explain more clearly which domain is bounded by the dotted line. Done.

P5 – Eqn 1. I don't think G and A are melt rates. What values were used for these parameters? The reviewer is right on the fact that they are not melt rates. Also a mistake was made when writing the equations: the actual form is $M_{b1} = H^\alpha (pG + (1-p)A)$ (The G parameter was changed so the equation looks like previous references to it in BISICLES related papers). The p parameter equals 1 at the grounding line and decreases exponentially away from it, as described in Cornford et al. (2015), and to which we refer to in the paper.

P6 – How does p vary? Give more details of exponential decay rate. p equals 1 at the grounding line and decreases exponentially as a distance to the grounding line to equal 0 away from it. We now refer to Cornford et al. (2015), Appendix B2, where the way p is computed is detailed.

P7 - Line 3. It needs to be clearer which equations are being solved. The text is currently ambiguous and leaves open three different possibilities. Is it (a) the L1L2 system described by Hindmarsh (2004), (b) the L1L2 system described by Schoof and Hindmarsh (2010) or (c) the SSA* system described by Cornford et al. (2015). If the equations are the SSA* approach described by Cornford et al (2015) then these are not the same as the

model described by Schoof and Hindmarsh (2010) and shouldn't be referred to as such.

The reviewer is right. The BISICLES model solves the SSA* approach described by Cornford et al. (2015). We clarified it in the text.

P7 - Eqn 5. Is any regularisation used in the sliding law for low velocities?

Yes. The results come from Berger et al 2016. Since this relates to a similar location, we simply re-used the C and Phi fields. This was not clear in the text and it's been clarified.

P8 – Imposing a trial and error value of basal freezing to prevent grounding line motion during relaxation seems slightly imprecise. An alternative approach is to fix the thickness of floating ice shelf to prevent grounding line migration during relaxation (see Arthern et al. 2015, DOI: 10.1002/2014JF003239). This approach can also incorporate dhdt observations on the grounded ice so the grounding line retreats at the observed rate during the forward simulation. It would be worth pointing out that alternative approaches to constraining the grounding line during surface relaxation are available.

Done.

P16 – Line 9. Capitalise Weertman.

Done.

Table 1. Check the units for heat capacity.

It was indeed wrong. We changed the units. We also corrected a typo in the value of $\gamma_t$.

Table 1. The parameter $\alpha 2$ is described as a tuning parameter for Mb2 not Mb1.

Right, changed.

**Response to Anonymous Referee 2**

1 General statement

The manuscript "Dynamic influence of pinning points on marine ice-sheets stability: a numerical study of Dronning Maud Land, East Antarctica" studies the impact of the presence of pinning points on outlet glaciers stability and their contribution to sea level rise. It shows that pinning points provide additional buttressing and therefore decrease the ice discharge by about 10%, but also that their presence strongly affects the ice shelf rheology inferred from data assimilation and therefore the model initial conditions. The authors suggest that including or omitting these pinning points impacts grounding line retreat and collapse of promontory on the timescale of several hundreds of years.

The manuscript is well written and the figures appropriate. The main point missing in the paper is that the authors do not show that their results are not resolution dependent. This can be easily done by rerunning a couple experiments with a higher mesh resolution and must be done in order to be confident that the results presented in this paper are robust.

The reviewer is perfectly right to ask for such a sensitivity study. This was also a major point raised by Robert Arthern, the other reviewer. Please look at the corresponding response above.

The interpretation of the impact of the melting is also quite ambiguous, as it differs in the discussion and conclusions. The section below describes in more details these two points and a few other specific comments.

See the responses to comments below.

2 Specific comments

p.1 l.16: "collapse" does not seem to be an appropriate word to describe a rather natural phenomenon that happens during simple grounding line retreat. This is also quite different from what is presented in the results.

The word "collapse" was indeed too strong. We now use "transition" instead.

p.2 l.11: The statement of ice rises not being detected by satellite observations is surprising: measurements of velocity and grounding line have a resolution of a few hundred meters while observations from altimetry also have an along track resolution a few hundred meters and the tracks are spaced by a few kilometers. Only bedrock topography does not have the required resolution, but sounding radars are operated in airborne and not satellite. So satellite observations, and in particular grounding line mapping should have the capability to resolve small ice rises and pinning points.

The reviewer is mostly correct and we modified the sentence accordingly.

p.2 l.34: The L-curve analysis is described in more details in Jay-Allemand et al. (2011) and not Gillet-Chaulet et al. (2012).
Agreed, we now refer to Jay-Allemand et al. (2011) instead of Gillet-Chaulet et al. (2012).

p.3 l.13: The eastern ice shelf of Thwaites Glacier experienced a complex behavior during the past coupled decades, with successive periods of acceleration and deceleration (Mouginot et al., 2014) that are not coherent with the gradual grounding line retreat and unpinning of the eastern shelf ice rise (Rignot et al., 2014). Entrainment of the Eastern ice shelf by the main ice shelf and changes in the region between the two parts of the ice shelf in this zone of intense shear is the preferred scenario to explain the complex changes observed and not a simple acceleration of this ice shelf (Mouginot et al., 2014).
The reviewer is right but what he mentions is actually related to what happened to the ice shelf before 2008. According to Mouginot et al. (2014): "After 2008, the TEIS accelerated again, but a restoration of the coupling between the two ice shelves seems unlikely as the main ice tongue calved in 2010. The recent acceleration might be better explained by a reduced buttressing of the pinning point at its terminus [Tinto and Bell, 2011; MacGregor et al., 2012] and/or the retreat of its grounding line due to enhanced thinning caused by warmer ocean water". We however rephrased, mentioning the eastern ice shelf only and not the whole ice shelf of Thwaites Glacier, to make the text clearer.

p.4 l.25-30: How sensitive are the results to these two additional excavations?
The excavation was made to avoid spurious re-advances of the grounding line that should not happen, and because the bed elevation beneath ice shelves was crudely interpolated by previous studies. For most low and several medium melting scenarios, the grounding line re-advance in the shallower water area upstream of Derwael Ice Rise. Using the original Bedmap2 bed elevation there would produce even larger re-advance, decreasing the sea level contribution for spurious reasons (which is written in the text). When the grounding line is stable or retreats, there is no sensitivity to excavations of course.

p.7 l.7: Why limit the resolution to 1km? BISICLES does not have problems using improved resolution and the domain simulated is small enough that increasing the resolution to 500 or 250 m should not too much or a problem. This is especially surprising at this manuscript focuses on the impact of small ice rises and pinning points. Authors would need to show that their results are not resolution dependent by performing a couple of the simulations that experience large changes with a grid resolution divided by two (500 m or less at the grounding line).
Done. See my answer to the first reviewer above.

p.7 l.16: How are the inversions of C and φ performed? Are they done simultaneously or one after the other? Are they done over the same region of different parts of the domain? This is not clear form the text and is an important question as changes in friction and stiffening factor can have a similar impact on ice flow.
The two inversions are performed simultaneously and over the whole domain, which we added in the text.

p.8. l.10: How large is the melt rate in this region? It is surprising to see that adding just 1 m/yr drastically change the grounding line evolution from rapid retreat to small advance. This suggests that the model is very sensitive to this parameter.
More precisely, the drainage basin of Hansenbreen in particular is very sensitive to this parameter, which is because the retrograde bed slope area (towards the ocean) starts few kilometres upstream of the current grounding line. To answer your question, we had a personal communication with Dr Depoorter (Depoorter et al., 2013) who recalculated the melting for the basins that we simulated. The sub-ice shelf melting undergone by the Hansenbreen glacier is about 3.5 Gt/yr. In the analogous study of Rignot et al. (2013), the overall sub-ice shelf melting is 7 Gt/yr but takes into account the two neighbouring glaciers to the west (called Borchgrevink in his paper). Our low, medium and high melt scenarios give more or less 2, 3.5 and 5 Gt/yr, respectively, after initialisation.

p.12 l.17-18: This statement contradicts what is shown on Fig.7. The type of melt rate applied seems to play a significant role in at least the rate of grounding line retreat as the three upper plots with melting type Mb1 all have a similar grounding line evolution, while the three lower ones with melting type Mb2 also have a similar behavior, distinct from the previous one. And this is actually quite different from what is summarized in the conclusions.
Yes, the two melt rates affect differently the buttressing from the ice shelf. We agree that the amount of retreat that can be attributed to MISI is difficult to assess. We therefore decided to add a Sup figure, comparable to Figure 6(b) for which we swith off the sub-ice shelf melting beneath the Hansen glacier once the grounding

line enters the retrograde bed slope area. This somehow enables to evaluate the effect of MISI and sub ice-shelf melting in the retreat.

3 Technical comments

p.2 l.6: "to more investigated" → "to be more investigated"
Done.

p.2 l.32: "overmatching" → "over fitting"
Done.

p.3 l.30: Which field measurements?
Radar and GPS: we added a reference to Drews (2015).

p.4 l.2: How do the first decades of the simulations compare with the observations that we have of the past couple decades?
The surface elevation change rates after the 50 years of relaxation are shown in Figure 2. These are comparable to other studies such as Cornford et al. (2015). In terms of velocities, the results remains similar to the initial velocities that were used to infer the $C$ and $\phi$ parameters. After the relaxation phase, the buttressing is modified with the new sub-ice shelf melting and significant changes start with the retreat of the Hansenbreen glacier, followed by much slower changes elsewhere.

p.4 l.3: How were these melt rates chosen? How do they compare with observations?
As said further in the text, we chose sub-ice shelf melt rates that are similar over the drainage basins to current values from Rignot et al. (2013) and Depoorter et al. (2013), with a personal communication from M. Depoorter who computed the sub-ice shelf melt rates for the specific basins of Hansenbreen and its two neighbours in the west, which were taken as one area in Rignot et al. (2013) under the name of Borchgrevink. These are for the medium melt rates scenarios. The low and high melt rates scenarios represent more or less half and twice the current values. We added a few words in the text and a Supplementary Table to summarize these numbers.

p.4 l.32: Authors should quickly summarize the model or observations that were used to derive this surface mass balance, and the year that is reproduces.
Done.
Fig. 2 caption: Notations should be consistent: B/S or Be /S
Done.

p.7 l.24-25: Not clear. Simply say that you use the improved velocity and the standard bedrock topography maps.
We clarified the text.

p.8 l.19: What are the values from Rignot et al. (2013) and Depoorter et al. (2013) and what are the values used in the simulation for the different ice shelves? A table with the melt observed and used for each ice shelf could help make this comparison.
We added a Supplementary Table to summarize current values given in Rignot et al. (2013) and given by M. Depoorter as personal communication, computed following the method given in Depoorter et al. (2013).

p.9 l.8: "mismatch" → "difference"
Done.

p.9 Fig. 3: This figure would be much clearer with colors.
Done.

p.11 Fig.5: Same as Fig.3, would be better with colors.
Done.

p.11 Fig.5 caption: "Velocity absolute" → "Absolute velocity"
Done.

p.12 l.7: There are many other very relevant references for the collapse of the WAIS.
Right, We added the following ones: Mercer (1978); Joughin and Alley (2011).

p.12 l.13: "In total" → "Overall"
Done.

p.16 l.9: Weertman, capital missing
Done.

**References**

[revised manuscript text omitted]

**Supplementary Figure** 2. sensitivity to grid resolution between 250 m and 4000 m at the grounding line, performed for the $B_e/S/L/M_{b1}/A_m$ experiment. (a) Cumulative sea level changes. The 4000 m resolution give the lowest changes while the 250 m resolution gives the highest changes. The differences between 250 m and 500 m are not significant. The 500 m resolution has therefore been chosen to run all the simulations discussed in the main paper. (b) to (f) Grounding-lines shown every 100 years (colorscale shown in (d)) for the resolution indicated at the top right of each panel.

[Figure]

**Supplementary Figure 3.** Ice velocity profiles shown every 100 years for the (a) $B_e/S/L/M_{b1}/A_m$ experiment along the flowline shown in Figure 6(a), and (b) for the same experiment in which the sub ice-shelf melting is switched off beneath the Hansenbreen ice shelf when its retreating grounding enters the upsloping part of the bed. The sub ice-shelf melting for other parts of the computational domain are not altered. The grounding line position is marked by the limit between solid and dashed parts of profiles.

**Supplementary Table.** Observed and initially modelled sub-ice shelf melt rates by drainage basins in Gt a$^{-1}$, from West to East within the computational domain (Figure 1). Our initial melt rates are comparable to what is given in Rignot et al. (2013) and by a Personal Communication (PC) from M. Depoorter (using the same technique as in Depoorter et al. (2013)). The exact numbers could not be reproduced for the medium melt-rates experiments since the sub-ice shelf melt rates are constrained by unique values over the computational domain. LIS: Lazarev ice shelf; UG: Unnamed Glacier; TB: Tussebreen; HB: Hansenbreen; RBIS: Roi Baudouin ice shelf.

| Glacier | | LIS | UG | TB | HB | RBIS | total |
|---|---|---|---|---|---|---|---|
| Observations | Depoorter et al. (2013) + PC | 1.8 | 0.5 | 3.6 | 3.7 | 20.2 | 29.8 |
| | Rignot et al. (2013) | 6.3 | \multicolumn{3}{c}{7.5 (UG+TB+HB)} | | 14.1 | 27.9 |
| Simulations | Low rates | 0.65±0.05 | 2.5±0.5 | 2.75±0.25 | 1.5±0.5 | 7±1 | 15±0.5 |
| | Medium rates | 1.4±0.2 | 4.5±0.4 | 4.9±0.1 | 2.5±0.5 | 14±2 | 28±2 |
| | High rates | 2.2±0.3 | 6.9±0.4 | 7.1±0.15 | 4.5±1.5 | 22±3 | 42±2 |